# Urine Untargeted Metabolomic Profiling Is Associated with the Dietary Pattern of Successful Aging among Malaysian Elderly

**DOI:** 10.3390/nu12102900

**Published:** 2020-09-23

**Authors:** Nik Nur Izzati Nik Mohd Fakhruddin, Suzana Shahar, Intan Safinar Ismail, Amalina Ahmad Azam, Nor Fadilah Rajab

**Affiliations:** 1Dietetic Programme, Centre for Healthy Aging and Wellness (H-CARE), Faculty of Health Sciences, Universiti Kebangsaan Malaysia, Jalan Raja Muda Abdul Aziz, Kuala Lumpur 50300, Malaysia; izzati.fakhruddin@gmail.com; 2Laboratory of Natural Products, Institute of Bioscience, Universiti Putra Malaysia, Serdang, Selangor 43400, Malaysia; safinar@upm.edu.my (I.S.I.); amalina_azam@hotmail.com (A.A.A.); 3Biomedical Science Programme, Centre for Healthy Ageing and Wellness (H-CARE), Faculty of Health Sciences, Universiti Kebangsaan Malaysia, Jalan Raja Muda Abdul Aziz, Kuala Lumpur 50300, Malaysia; nfadilah@ukm.edu.my

**Keywords:** older adult, metabolomic profiling, ^1^H-NMR, successful aging, biomarker, principal component analysis, partial least-squares discriminant analysis

## Abstract

Food intake biomarkers (FIBs) can reflect the intake of specific foods or dietary patterns (DP). DP for successful aging (SA) has been widely studied. However, the relationship between SA and DP characterized by FIBs still needs further exploration as the candidate markers are scarce. Thus, 1H-nuclear magnetic resonance (^1^H-NMR)-based urine metabolomics profiling was conducted to identify potential metabolites which can act as specific markers representing DP for SA. Urine sample of nine subjects from each three aging groups, SA, usual aging (UA), and mild cognitive impairment (MCI), were analyzed using the ^1^H-NMR metabolomic approach. Principal components analysis (PCA) and partial least-squares discriminant analysis (PLS-DA) were applied. The association between SA urinary metabolites and its DP was assessed using the Pearson’s correlation analysis. The urine of SA subjects was characterized by the greater excretion of citrate, taurine, hypotaurine, serotonin, and melatonin as compared to UA and MCI. These urinary metabolites were associated with alteration in “taurine and hypotaurine metabolism” and “tryptophan metabolism” in SA elderly. Urinary serotonin (r = 0.48, *p* < 0.05) and melatonin (r = 0.47, *p* < 0.05) were associated with oat intake. These findings demonstrate that a metabolomic approach may be useful for correlating DP with SA urinary metabolites and for further understanding of SA development.

## 1. Introduction

Metabolomics is a novel scientific discipline focused on the association between disease and metabolic profile in tissue and biofluids, as determined by techniques including ^1^H-nuclear magnetic resonance (^1^H-NMR) spectroscopy and mass spectroscopy. Untargeted metabolomics is an “omic” approach that provides comprehensive assessment of various metabolite changes that provides information on altered metabolism of metabolic pathways resulting from their biological system effects [1]. Several pathways are known to be involved in the mechanism of aging, such as free radical production, lipid peroxidation, and inflammatory response, all associated with metabolic changes [2,3,4]. Metabolomic profiling can be easily performed using peripheral tissue, cerebrospinal fluid, plasma, and urine, making this approach valuable in clinical applications. The profile of metabolites using serum, plasma, saliva, and cerebrospinal fluid in relation to mild cognitive impairment, dementia, and Alzheimer’s disease has been widely performed [1,5,6,7,8,9]. There are metabolomic studies related to health span and biological aging in human [10,11,12]. Amino acid metabolites and lipid associated with health span indicator and rate of biological aging. Higher concentration of plasma metabolites from TCA cycle, i.e., citrate and tryptophan metabolism were associated with lower biological age [10,12]. However, the metabolites that are likely to be biomarkers for successful aging (SA) remain unknown. Metabolomics offers a potential in terms of nutritional evaluation as it measures various small molecules present in biological systems. When combined with multivariate statistics analysis, it is able to characterize the physiological conditions of several different biofluids not only in clinical practice, but also in developing hypotheses towards SA.

Our previous study, which showed dietary patterns (DP) of “oats and tropical fruits”, increased 2.4 times chances of SA among those with secondary education and above [13]. Other than that, Hodge et al. (2014) suggested a DP including plenty of fruits while limiting meat and fried foods may improve the likelihood of ageing successfully [14]. Untargeted metabolomics can provide a clearer understanding of how humans respond to complex diets [15]. By conducting a simultaneous analysis of the metabolites contained in the urine, an individual’s level of adherence to the DP studied can be identified [16]. The combination of metabolic biology will produce a better and holistic analysis of DPs compared to the use of dietary history questionnaires alone. With the sophistication of metabolomic technology, small molecules of metabolites can be detected, measured, and correlated with specific DPs. Metabolite profiles may play a role in the nutritional evaluation and determination of food intake biomarkers (FIBs). This study is important in looking at the FIBs of DP in which the metabolomic profile is associated with the dietary groups of different aging groups. To date, studies examining between metabolomic and DP have been conducted [17,18], but unfortunately, no specific studies based on FIB of SA groups have been identified.

## 2. Materials and Methods

### 2.1. Study Population and Study Design

This is a comparative study, as part of a larger population based on a longitudinal study, which investigates neuroprotection model for the purpose of healthy longevity among Malaysian elderly, namely towards useful aging (TUA). The methodology of this study was described earlier [19]. Briefly, subjects were chosen using multi-stage random sampling method from states with the highest prevalence (at least 10% from the total population of the state) of elderly according to each zone in Peninsular Malaysia; Perak (North), Selangor (Central), Kelantan (East), and Johor (South). This study was divided into three phases. Phase 1 and 2 have been described earlier [13,20]. The study protocol was approved by the Research and Medical Research Ethics Committee of University Kebangsaan Malaysia (UKM), and informed consent was obtained from all participants.

### 2.2. Criteria of Successful Aging

At baseline, participants were divided into three different aging groups, namely mild cognitive impairment (MCI), SA, and usual aging (UA). The classification of aging groups was based on cognitive functions, physical functions, and existing six chronic diseases, including diabetes mellitus, hypertension, stroke, cancer, chronic obstructive pulmonary disease, and heart diseases, as well as quality of life and self-health rated according to Shahar et al. (2015) [19]. These are described in Table 1. Cognitive functions were tested using the mini mental state examination (MMSE) [21], digit span [22], and Rey auditory verbal learning test (RAVLT) [23]. Physical functions were assessed by using activity of daily living (ADL) [24] and instrumental activity of daily living (IADL) [25]. Two psychosocial questions were asked regarding quality of life and self-perceived health. This phase involved a total of 27 subjects selected and matched for age and aging groups from previous analysis [13].

### 2.3. Identification of Dietary Pattern for Successful Aging

DP of the elderly in Malaysia was identified in Phase 2 of this study [13]. Dietary intake was obtained from subjects or caregivers by using a validated dietary history questionnaire [20]. Then, food items recorded in DHQ were extracted and classified into 14 food groups based on their similarities or references from other studies. Principal component analysis (PCA) was applied to identify the dietary patterns of the elderly in Malaysia using Statistical Package for Social Sciences (SPSS) IBM 21 to derive DPs on the basis of the original food group variables (consumption in g/day). 

DP was labeled based on food groups that exhibited the strongest correlation by having the highest loading factor. Food groups were retained in the DP if the factor loading value was >0.3. DP score for each subject was then calculated by adding the intake of 14 food groups weighted by their factor loading. 

Ordinal logistic regression was used to assess the association between the DPs and SA with adjustments of age, gender, race, calorie intake, body mass index (BMI), marital status, and smoking status as covariates for Model 1. Meanwhile, Model 2 included educational level, in addition to other covariates mentioned in Model 1.

### 2.4. Sample Preparation

Urine samples were collected in 60-mL urine collection bottles for metabolomic analysis. As the first void urine contained more variables than the subsequent voids, the second urine was used in this study [26]. Urine samples were centrifuged for 3000 rpm for 10 min. Then, urine samples were transferred into multiple aliquots of 1.5-mL sterile microcentrifuge tubes. The microcentrifuge tubes were sealed tightly and stored at −80 °C until being used for metabolomic analysis [19]. 

The preparation of urine samples for metabolomic analysis by ^1^H-NMR was performed manually [27]. After being thawed at room temperature, 400 μL of the urine sample was then mixed with 200 μL of 0.3 M sodium phosphate buffer (1 mM TSP (sodium trimethylsilyl [2,3,3,3-2H4]propionate) and 20% D_2_O) (pH 7.4) in 1.5 mL microcentrifuge tubes, and later centrifuged at 9600 rpm for five minutes at 4 °C. A total of 550 μL of the sample was transferred into a 5-mm NMR tube. The TSP acted as internal chemical shift reference, and D_2_O as lock signal for the NMR spectrometer.

### 2.5. NMR Acquisition

The NMR spectra were acquired using a Bruker Ascend 600 MHz NMR Spectrometer (Bruker Biospin, Rheinstetten, Germany). The NMR experiments were obtained at the temperature of 26.85 °C. Before measurement, each sample was loaded into the prop and the temperature was calibrated and kept constant for 3 to 5 min. In order to observe the dynamic range of metabolites concentrations efficiently, the water signal was suppressed by running 1D nuclear Overhauser enhancement spectrometers (NOESY)-presat experiments. Standard one-dimensional (1D) ^1^H-NMR spectra were acquired using a single 90° pulse length experiment with water presaturation using a relaxation delay of 2 s. Each data set was averaged over 64 transients using 32,768 time domain points and 64 scans. The data were Fourier transformed, and spectra were referenced to the TSP signal at 0 ppm.

Each NMR spectrum baseline correction, phasing, and chemical shift calibration was done. Then, all spectrums were binned to 0.04 ppm wide segments between 0.00 and 10.0 ppm. All pre-processing steps were completed using Chenomx NMR Suite Professional software version 8.3 (Chenomx Inc., Edmonton, AB, Canada), giving a total of 219 integrated regions per NMR spectrum. The spectrum regions of water (δ = 4.8 ppm) and urea (δ = 5.8 ppm) were removed from the analysis for all groups in order to prevent baseline effects of variability in the suppression of the water resonance and the nonquantitative contribution of urea [27,28]. Each NMR variable was normalized to the total area under peak curves in order to allow a spectrum-to-spectrum comparison.

### 2.6. Urinary Metabolites Identification

Urinary metabolites were assigned by comparing spectrum obtained from NMR with the chemical shifts and peak shapes of standard compounds from internal database of reference spectra in Chenomx Profiler software version 8.3 (Chenomx Inc., Edmonton, AB, Canada) [29,30]. These metabolites were then compared with NMR spectral data, i.e., those available in Human Metabolome Database (http://www.hmdb.ca) and along with the existing NMR-based metabolomic literature [31]. 

### 2.7. Statistical Analysis

The bucketed spectral data were converted to Microsoft Excel format and imported into SIMCA-P+ software version 13.0 (Umetrics, Umeå, Sweden) for multivariate analysis in which all spectral data were transformed into Log-10 mode, mean-centered with UV scaling [18,32]. Initially, principal component analysis (PCA) was applied to identify the possible outliers of the clustering. Partial least squares-discrimination analysis (PLS-DA) was then advanced for pattern recognition analysis between aging groups and facilitated the detection of metabolites consistently present in the urine sample [33]. All the statistical analyses were carried out using binned bucket data and the identified metabolites for spectral binning. PLS-DA made it possible to rotate the projection to latent variables that focused on class separation. Thus, PLS-DA aimed to find a model that separated classes of observations based on their x variables. The PLS-DA model was validated by a permutation method by describing R2Y and Q2Y values and CV-ANOVA (expressed as *p*-values for the model) [34]. In addition, the choice of input variables for univariate analysis was based on magnitude of the variable influence on projection (VIP) from SIMCA-P where a VIP >1 indicated significant contribution to the model [9].

A Student’s *t*-test was carried out to observe the metabolites that showed a significant relative intensity difference between two aging groups (e.g., SA vs. UA; SA vs. MCI and UA vs. MCI) [35]. The spectral data, created for the multivariate analysis, was used as an input for Student’s *t*-test variable. For each bucket data, the corresponding *p*-valued was calculated. In order to address false discovery rate (FDR) from multiple comparison, Benjamini–Hocherberg correction (0.10) was applied on each pair of analysis [10,36].

In Phase 2, the DP with model, adjusted to age, gender, race, smoking status, BMI, energy intake, and marital status, showed positive association with SA. The relationship between DP scores for SA, “oats and tropical fruits” (Phase 2) with SA urinary metabolites, was performed using Pearson correlation. Thus, this model was also adjusted to age, gender, race, smoking status, body mass index, energy intake, marital status, and education level.

### 2.8. Pathway Analysis

For the probable perturbed pathway in looking into the mechanism of actions, the Metaboanalyst 4.0 (https://www.metaboanalyst.ca) was utilized. Identified urinary metabolites were entered in a column for further analysis, and had to be matched with HMDB database. The hypergeometric test was chosen for over representation analysis and relative-betweenness centrality for pathway topology analysis. For the pathway library, *Homo sapiens* was chosen.

## 3. Results

### 3.1. Urinary Metabolites Identification

In this study, we analyzed human urine from three aging groups, namely SA, MCI, and UA using ^1^H-NMR spectroscopy. NMR spectra for all aging groups showed a consistent set of metabolite signals present. NMR spectra of urines were binned into individual spectra bins accounting for one or more multiplets. The number of spectra bins in each urine was 219, of which 72 bins were assigned to 23 urinary metabolites. Identified urinary metabolites are summarized in Table 2 and Figure 1, and Appendix A displays the representatives of ^1^H-NMR spectrum of each aging group.

### 3.2. Principal Component Analysis (PCA) of the Urine Samples 

The mean-centered ^1^H NMR data were imported into the SIMCA-P software and visualized by the unsupervised model of PCA (Figure 2). *R*^2^ values of PCA model may be used to assess its degree of fit to the data [37]. PCA degree of fit will return only *R*^2^*X*, to which the principal components describe the observation data (Appendix A). PCA score plots showed the clusters of urine samples in two principal components of PC1 (t(1)) and PC2 (t(2)) which accounted for 50.3% and 12.5% data variations, respectively. The diverged score plots signified a separation between SA and UA with MCI (negative quadrant) by PC1, with SA and UA assembled together in one cluster (positive quadrant). For further elucidation according to the aging groups and providing validation, supervised analysis of the pattern recognition, such as PLS-DA, was performed.

### 3.3. Partial Least Squares-Discriminant Analysis (PLS-DA) of the Urine Samples 

The spectral data of the urine samples from the SA (*n* = 9), UA group (*n* = 9), and MCI (*n* = 9) in PLS-DA scores and loading plots are depicted in Figure 3 and Figure 4, respectively. A similar cluster pattern was obtained for the PLS-DA model as observed in PCA (Figure 2). The UA and SA were clearly separated from MCI by PC1 (t(1)). SA and UA seemed to give two clusters looking by PC2 (t(2)). However, there were some variables belonging to these groups that shared a close similarity shown by their proximity in the middle part of the principal component (t(2)). 

Figure 4 indicates the distribution of urinary metabolites based on the variations due to perturbation occurred. These metabolites were chosen as their variables importance projection (VIP) values > 1 (Appendix A), which were considered to be influential for clustering of the groups in the PLS-DA model [9]. Sixteen urinary metabolites were identified as the key compounds in the classification of the groups in which all of them were detected in SA and UA. Hence, this observation suggests that MCI experienced the lower or loss of these sixteen urinary metabolites compared to SA and UA. 

Among the sixteen urinary metabolites as listed in Figure 4 and Table 2, six were identified as significantly influenced by the variations as their bars did not cross zero [38] as shown in the loading column plot (Figure 5). Corresponding to the PLS-DA loading plot in Figure 4, the six significant urinary metabolites, namely (4) melatonin, (7) succinate, (8) citrate, (9) hypotaurine, (10) taurine, and (12) serotonin, were all located in SA seen by PC2 (w*c(2)). This suggests that SA experienced notable and slight changes of the six urinary metabolites in comparison to the MCI and UA, respectively.

### 3.4. Validation of PLS-DA Model

The quality of the principal component models was evaluated with parameters R2 and Q2. The goodness of fit was quantified by R2 while the predictive ability was indicated by Q2. Generally, R2Y and Q2Y explain variation in the data, wherein 0 shows no explained variation, and 1 is 100% variation accounted, while above 0.5 of both R2 and Q2 satisfy for a qualified model [6,9]. The current PLS-DA model indicated R2Y = 0.64 and Q2 Y = 0.50, which suggested good predictability and fit of the used model in metabolic discrimination between the aging groups. 

The permutation test was performed to further validate the PLS-DA model [34]. It provided the statistical significance of the estimated predicted power of the models by comparing R2Y and Q2Y values of the original model with those of the reordered model, which was created newly whenever y data was permutated at random. Models with the R2Y intercept < 0.4 and Q2Y intercept < 0.05 indicate as validated [34]. As shown in Figure 6, the PLS-DA model gave satisfactory values of R2Y intercept = 0.308 and a Q2Y intercept = −0.0801. Additional CV-ANOVA value of *p* < 0.001 further confirmed the validity of the model [34].

### 3.5. Relative Quantification of Identified Urinary Metabolites

For further explanation on the perturbed putative urinary metabolites, an unpaired *t*-test for quantification of the selected urinary metabolites between two groups was performed. The mean for all urinary metabolites’ concentrations except hippurate in SA was determined to be higher than MCI (*p* < 0.05), while all urinary metabolites statistically significantly greater in UA compared to MCI (*p* < 0.05) except serotonin and citrate. Only serotonin, melatonin, taurine, hypotaurine, and citrate were higher in SA as compared to UA (Table 3).

### 3.6. Probable Metabolic Pathways for Successful Aging (SA)

MetaboAnalyst 4.0 was utilized to extrapolate the possible disturbed metabolic pathways based on the five significant changed urinary metabolites resulted from the multivariate analysis. Six pathways might involve in major metabolic pathways: (1) Taurine and hypotaurine metabolism (*p* = 2.31 × 10^−4^, impact = 0.714); (2) tryptophan metabolism (*p* = 0.01, impact = 0.132); (3) citrate cycle (*p* = 0.06, impact = 0.09); (4) alanine, aspartate, and glutamate metabolism (*p* = 0.09, impact: 0.000); (5) glyoxylate and dicarboxylate metabolism (*p* = 0.10, impact 0.032); and (6) primary bile acid biosynthesis (*p* = 0.14, impact 0.007 (Figure 7). Two biochemical pathways achieved statistical significant (*p* < 0.05) and exhibited a “pathway impact” score > 0.1 [33]. These two metabolic pathways were “taurine and hypotaurine metabolism” and “tryptophan metabolism”, associated with the amino acid metabolism. 

### 3.7. Relationship between Dietary Pattern and SA

Table 4 shows the relationship between SA urinary metabolites and “oats and tropical fruits” DP score and food groups. DP “oats and tropical fruits” was obtained by factor analysis from Phase 2. The DP score and tropical fruit food group did not show any relationship with the selective key urinary metabolites for SA, as all the *p*-values are above 0.05. However, only oat food groups showed significantly positive relationship with melatonin (r = 0.47, *p* < 0.05) and serotonin (0.48, *p* < 0.05).

## 4. Discussion

The main advantage of “omics” technologies is that there is no preselection of candidate metabolites to be investigated for a potential influence on the disorder under study. In particular, NMR-based metabolomics is a robust technique that does not require sample treatment, chromatography, or analyte ionization, and allows unambiguous identification of analytes. NMR was used here to approach the nonbiased, nontargeted study of a potential association of successful aging with a panel of urinary metabolites. The urine metabolomics platform was recently featured in a large scale AD and dementia metabolic profiling study [39,40]. In this study, the 1H-NMR-based metabolomic approach, involving three aging groups namely SA, UA, and MCI, successfully identified the major metabolic variations between the groups. Five urinary metabolites, namely (4) melatonin, (8) citrate, (9) hypotaurine, (10) taurine, and (12) serotonin were successfully concentratively detected in SA based on the PLS-DA model analysis and quantified by the univariate *t*-test. In relation to the identified compounds, two amino acid pathways of taurine and hypotaurine metabolism and tryptophan metabolism were proposed to be significant and probable mechanistic routes.

These findings are in accordance with studies done by Wang et al. (2014) and Zheng et al. (2012) [6,9], whereby the concentrations of taurine and hypotaurine were found lower in MCI as compared to the control subjects. The control subjects in the studies were comparable to SA of the present study as both had normal cognitive functions and not demented. Taurine and hypotaurine metabolism have been associated with neuroprotective where taurine acts as inhibitory amino acid neurotransmitter that gives protection against glutamate excitotoxicity [41,42]. The targeted metabolomic analysis significantly revealed lower plasma level of serotonin, phenylalanine, proline, lysine, phosphatidylcholine, taurine, and acylcarnitine which are associated with conversion of MCI to Alzheimer’s disease [43]. Glutamate excitotoxicity is found to cause neuronal injury and leads to neurodegenerative diseases, such as Alzheimer and Parkinson [41]. In addition, when a neuron is stimulated by the presence of glutamate, the concentration levels of Ca2+ increased due to the influx of calcium from extracellular through various channels. Taurine acts to reduce the Ca2+ concentration levels induced by glutamate via L-type, P/Q-, N-guided calcium channel voltage (voltage-gated calcium channel) and *N*-methyl-d-aspartate (NMDA), thus preventing membrane glutamate depolarization [42,44]. The effects of intracellular calcium regulation is to preserve the mitochondria from damage due to excess calcium which in turn causes cell death [45]. However, higher taurine level is also found in post-mortem brain tissue among Alzheimer patients as compared to control [46], which means the inconsistent taurine regulation is proof of possible impact on brain osmoregulation. Higher plasma levels of anthranilic acid and glutamate, and lower levels of taurine are associated with cognitive decline and the incidence or progression of dementia via the tryptophan-kynurenine pathway, which is also linked to glutamate excitotoxicity in the pathogenesis of dementia [47].

Tryptophan metabolism might also play an important role for SA. This study found that tryptophan urine levels were significantly different between SA and UA with MCI, but did not show any significant differences between each other. These are consistent with those reported by Graham et al. (2015) and Trushina et al. (2013) in which tryptophan level in plasma and cerebrospinal fluid among MCI and Alzheimer subjects were lower than in controls [1,5]. It is possible that the subjects in MCI group received less tryptophan from the dietary sources than SA and UA. In an earlier study, we found that SA group consumed more oats and tropical fruits [13] by which the former is more likely to be taken with tryptophan rich food sources, such as dairy products. When tryptophan sources are low in diet, serotonin biosynthesis in the peripheral nervous system and spinal cord systems would decrease [48]. Therefore, MCI has the lowest serotonin level followed by UA and SA. High intakes of tryptophan food sources could increase the availability of tryptophan in the blood and affect enzymes in the liver and blood. As a result, the concentration of tryptophan metabolites, serotonin, and kynurenine increased [49]. 

Based on DP, it was SA group who consumed more oats, i.e., the rich source of antioxidant, which may increase the absorption of tryptophan into the brain and reduced its damage. Dietary antioxidants help in serotonin biosynthesis through two ways of preserving tetrahydrobiopterin, a cofactor to the tryptophan enzyme 5-hydrolase from oxidation, and subsequently helps serotonin biosynthesis. In addition, dietary antioxidants have diligently showed an increase in tryptophan and L1 transporter to the brain by reducing the production of interferon-γ (IFN-γ) Th-1 and indolamine 2,3-dioxigenase (IDO) cytokines [49].

Melatonin and serotonin were positively correlated with the “oat food group”. As mentioned earlier, oats taken with milk is believed to cause the increase of tryptophan within the blood, thus affects enzymes in the liver [50]. As a result, the density of tryptophan metabolites, including serotonin, increases [49]. SA gave the highest serotonin and melatonin levels followed by the UA and MCI groups. According to Meng et al. (2017), SA individuals who consumed diet rich in fishes, eggs, legumes, cereals, sprouted nuts, and oats showed high melatonin levels [51]. Consumption of these foods suggests a greater concentration of melatonin in serum and its antioxidant capacity, which gives a good effect on health [51,52]. Melatonin production is lower during the day and increases at night because it is controlled by light or dark conditions [51]. Decreased melatonin plasma production at night causes circadian cycles to disrupt the sleep cycle. Sleep duration for about 6 to 8 h a day provides good health to older people [53]. SA subjects are likely to have a normal night-time sleep causing melatonin to be produced well, thus providing good self-health perceived. However, further studies need to be done to confirm this claim as this study did not investigate the sleeping pattern of the subjects.

The strength of this study is in its ability to correlate DP identified earlier among SA with specific metabolites excreted in the urine. Such a finding provides firmer evidence for public health policy and strategy as compared to DP outcome alone. Findings of this study lead to further analysis on other lifestyle components, such as physical activity, smoking, and alcohol intake with metabolites of SA. There is also a need to investigate the impact of the markers on longevity associated outcomes, such as multimorbidity, mortality, frailty, and dementia. The major limitation of the study is the small number of sample size. Even though untargeted profiling provides advantage for novel target discovery, there are difficulties in identifying and characterizing unknown small molecules. Thus, targeted profiling which convert spectral information to metabolites concentration before classification and using is recommended in the future [54,55]. Data normalization needs to ensure that a measured concentration or a fold change in concentration observed for a metabolite at the lower end of the dynamic range is as reliable as it is for a metabolite at the upper end [56]. The differences in renal function between aging group might affect the concentration of metabolites detected in urine. However, the data analyzed in this study were not normalized to the renal profile of the subjects and all the metabolomic data has been analyzed using an appropriate statistical analysis as referred to the relevant references accordingly. 

In addition, the NMR spectroscopy approach used only one-dimensional analysis of proton (1D) NMR data and provided limited interpretations of the metabolites. Therefore, two-dimensional (2D) analysis such as J-Resolved or heteronuclear multiple bond correlation spectroscopy (HMBC) and heteronuclear single quantum coherence (HSQC) are proposed to explore the metabolites involved in metabolic pathways with higher confidence level. In order to be able to identify meaningful FIBs study and get better outcomes in metabolomic analysis, a larger sample size, prospective studies and use of other biofluids such as plasma or serum might are needed to evaluate a causal link between DP and the biomarkers of SA [57].

## 5. Conclusions

Putative urinary metabolites associated with SA were identified as taurine, hypotaurine, serotonin, melatonin, and citrate; with tryptophan metabolism and taurine and hypotaurine metabolism. In addition, serotonin and melatonin showed positive correlations with oat consumption suggested to very well represent FIBs. The combination of dietary pattern and metabolomic profiles led to identification of FIBs of oat intake from “oats and tropical fruits” DP. Future undertakings of longitudinal impact of the biomarkers or metabolites on longevity outcomes seem to be essential.

## Figures and Tables

**Figure 1 nutrients-12-02900-f001:**
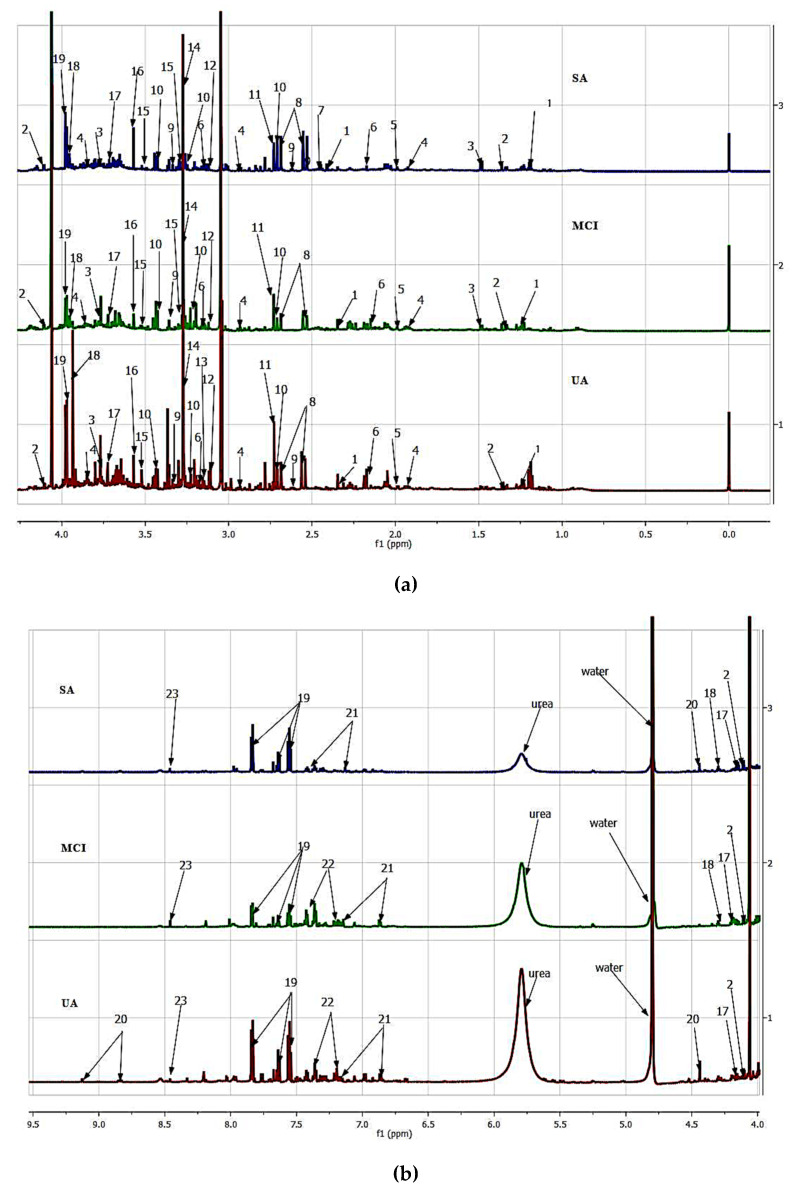
^1^H-NMR spectra (600 MHz) of urine in PBS for successful aging (SA), mild cognitive impairment (MCI), and usual aging (UA) representatives. (**a**) 0.0 to 4.0 ppm (**b**) 4.0 to 9.5 ppm. Putative identified metabolites for urine: (1) 3-Hydroxyisovalerate, (2) Lactate, (3) Alanine, (4) Melatonin, (5) Acetate, (6) *O*-acetylcarnitine, (7) Succinate, (8) Citrate, (9) Hypotaurine, (10) Taurine, (11) Malate, (12) Serotonin, (13) Dimethylsulphone, (14) Trimethylamine-*N*-oxide, (15) Caffein, (16) Glycine, (17) Kynurenine, (18) Galactarate, (19) Hippurate, (20) Trigonelline, (21) 4-Hydroxyphenyllactate, (22) Tryptophan, (23) Formate.

**Figure 2 nutrients-12-02900-f002:**
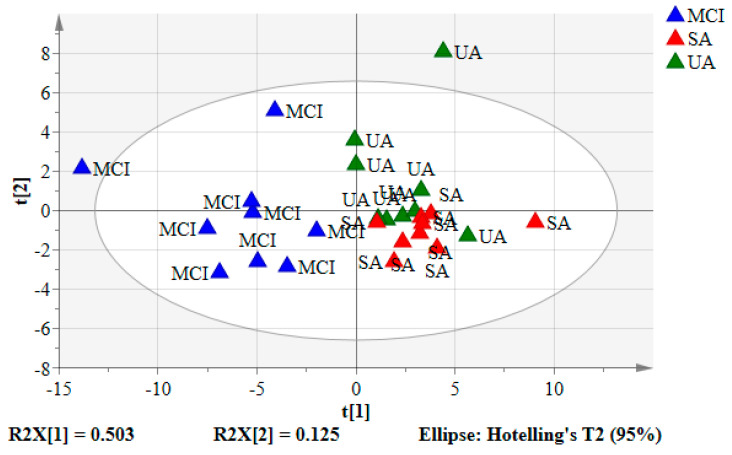
The PCA score plot of SA, UA, and MCI groups. Each point represented a single individual urine sample of the ^1^H-NMR spectrum. Blue triangles: MCI, red triangles: SA, and green triangles: UA.

**Figure 3 nutrients-12-02900-f003:**
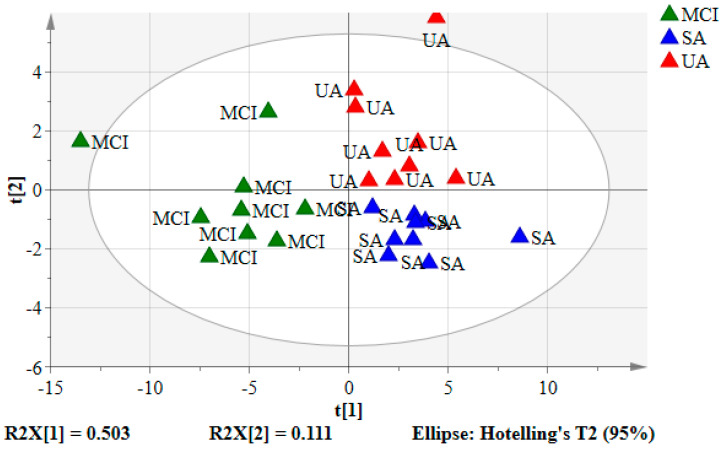
The PLS-DA score plot for SA, UA, and MCI groups. Each point represents ^1^H-NMR data of an individual urine sample. Blue triangles: SA, red triangles: UA, and green triangles: MCI.

**Figure 4 nutrients-12-02900-f004:**
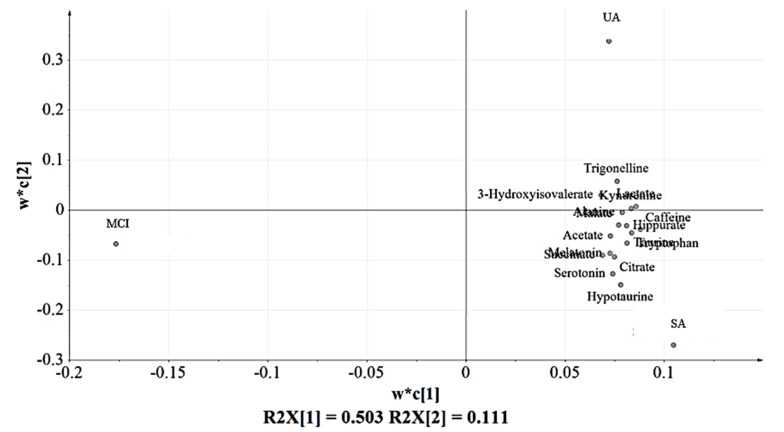
The PLS-DA loading plot with the putatively identified urinary metabolites mostly found in the positive side of PC1 (w*c(1)) corresponding to UA and SA of the score plot (Figure 3).

**Figure 5 nutrients-12-02900-f005:**
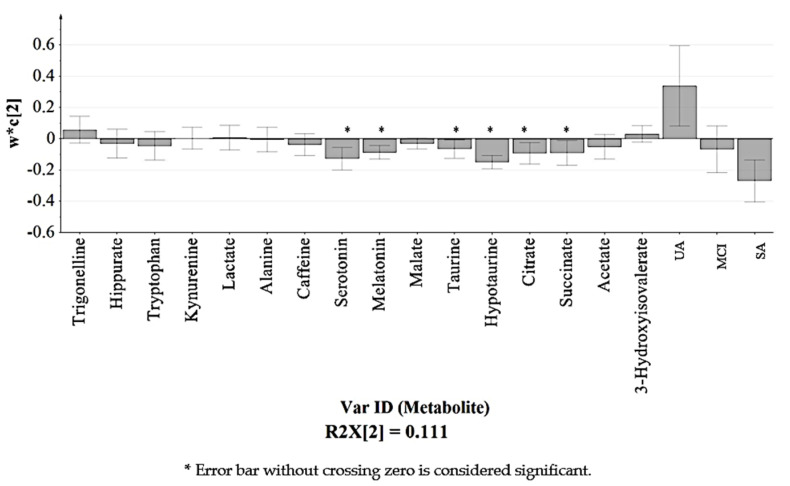
Loading column plot of PLS-DA model (Figure 4) observed by PC 2 (w*c(2)). Each column represents the putative identified urinary metabolite with standard errors displayed in error bar.

**Figure 6 nutrients-12-02900-f006:**
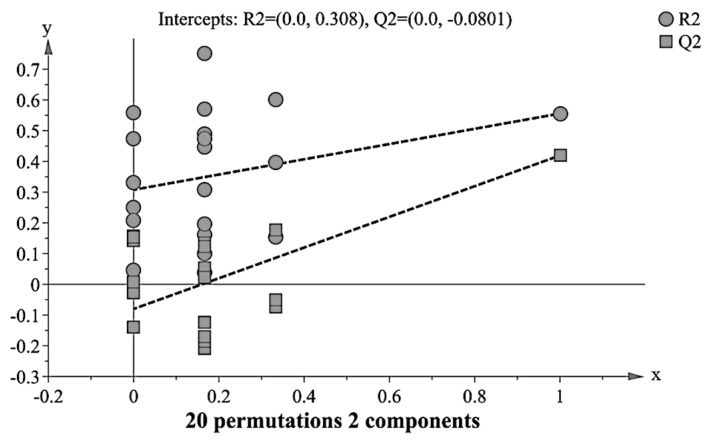
Model validation assessed by 20 permutation tests (R2 = (0.0, 0.308) Q2 = (0.0, −0.0801).

**Figure 7 nutrients-12-02900-f007:**
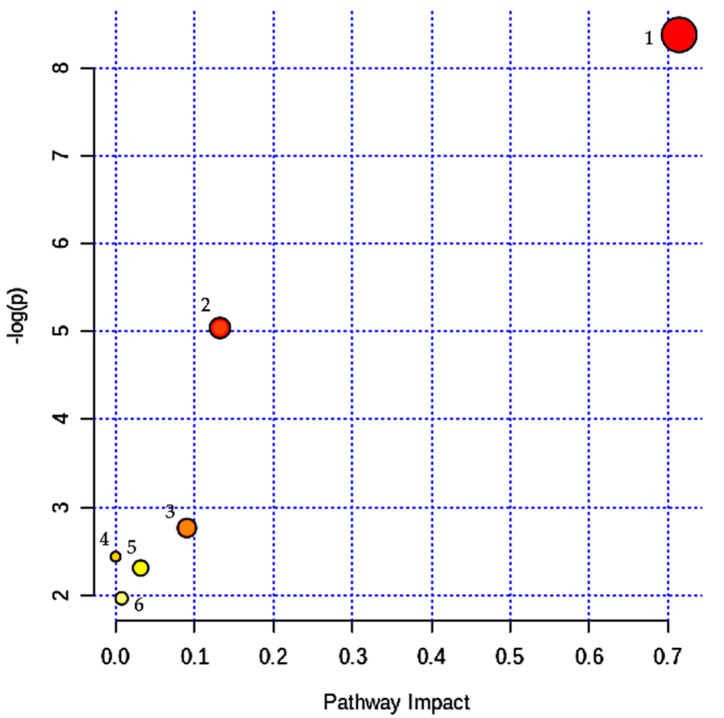
Summary of pathways analysis with MetaboAnalyst 4.0 (18 April 2020): (1) Taurine and hypotaurine metabolism, (2) Tryptophan metabolism, (3) Citrate cycle, (4) Alanine, aspartate, and glutamate metabolism, (5) Glyoxylate and dicarboxylate metabolism, (6) Primary bile acid biosynthesis.

**Table 1 nutrients-12-02900-t001:** Classification of aging groups [19].

Successful Cognitive Aging	Usual Aging	Mild Cognitive Impairment
Absence of six chronic diseases, including diabetes mellitus, hypertension, cancer, heart diseases, chronic lung diseases, and stroke	Scored above MCI and below SA in most of the cognitive test administered	Subjective memory complaint
No limitations in functional status with full score for both Activities of Daily Living and Instrumental Activities of Daily Living	Not demented	Objective memory impairment determined by Digit Span and Rey Auditory Verbal Learning Test with score at least 1.5 SD below the mean average
Normal global function with Mini Mental Examination State (MMSE score ≥ 22)	No or minimal in functional status	Not demented
Not depressed with Geriatric Depression Scale-15 score ≤ 4		No limitations in basic activities of daily living
Good quality of life		No or very minimal in instrumental activities of daily living by having a score of ≤1.5 SD from mean norm
Good self-perceived health		Global function by having MMSE score of ≥19

**Table 2 nutrients-12-02900-t002:** 1H-nuclear magnetic resonance (^1^H-NMR) data for the putative identified metabolites.

Key	HMDB ID	Urinary Metabolites	Source	Chemical Shift (Multiplicity, J Value)
1.	HMDB0000754	3-Hydroxyisovalerate	Endogenous and food	1.26 (s), 2.36 (s)
2.	HMDB0001311	Lactate	Endogenous and food	1.32 (d, 6.9), 4.11 (q, 6.9)
3.	HMDB0000161	Alanine	Endogenous and food	1.44 (d, 7.1), 3.79 (q, 7.2)
4.	HMDB0001389	Melatonin	Endogenous and food	1.91 (s), 2.93 (t, 6.8), 3.87 (s)
5.	HMDB0000042	Acetate	Endogenous and food	1.93 (s)
6.	HMDB0000201	*O*-acetylcarnitine	Endogenous and food	2.13 (s), 3.18 (s)
7.	HMDB0000254	Succinate	Endogenous and food	2.40 (s)
8.	HMDB0000094	Citrate	Endogenous and food	2.54 (d), 2.68 (d, 15.2)
9.	HMDB0000965	Hypotaurine	Endogenous	2.60 (t, 6.9), 3.32 (t, 6.9)
10.	HMDB0000251	Taurine	Endogenous and food	2.69 (s), 3.23 (t, 6.6), 3.42 (t, 6.6)
11.	HMDB0000156	Malate	Endogenous and food	2.73 (dd, 15.4)
12.	HMDB0000259	Serotonin	Endogenous and food	3.11 (t, 7.1), 7.41 (d, 8.7)
13.	HMDB0004983	Dimethyl sulfone	Endogenous and food	3.14 (s)
14.	HMDB0000925	Trimethylamine-*N*-oxide	Endogenous	3.26 (s)
15.	HMDB0001964	Caffein	Endogenous and food	3.29 (s), 3.50 (s)
16.	HMDB0000123	Glycine	Endogenous and food	3.55 (s)
17.	HMDB0000684	Kynurenine	Endogenous and food	3.73 (d), 4.14 (t, 6.5, 4.2)
18.	HMDB0000639	Galactarate	Endogenous and food	3.95 (s), 4.24 (s)
19.	HMDB0000714	Hippurate	Endogenous and food	3.96 (d, 5.8), 7.54 (m), 7.62 (tt, 7.5, 1.5), 7.82 (dd, 8.4, 1.2)
20.	HMDB0000875	Trigonelline	Endogenous and food	4.42 (s), 8.83 (m), 9.11 (s)
21.	HMDB0000755	4-Hydroxyphenyllactate	Endogenous	6.85 (d, 8.3), 7.15 (d, 8.2)
22.	HMDB30396	Tryptophan	Endogenous and food	7.18 (d), 7.29 (s)
23.	HMDB0000142	Formate	Endogenous and food	8.44 (s)

s = singlet, d = doublet, dd= doublet of doublets, t = triplet, tt= triplet of triplets q = quadruplet, m = multiplet.

**Table 3 nutrients-12-02900-t003:** Relative quantification of the discriminatory urinary metabolites identified from the PLS-DA analysis of the ^1^H-NMR spectra aging groups, based on the mean peak area of the ^1^H NMR signals.

	Chemical Shift (ppm)	Changes		
		SA vs. UA	SA vs. MCI	UA vs. MCI
3-hydroxyisovalerate	1.26	−	+ *	+ *
Acetate	1.93	+	+ *	+ *
Malate	2.73	+	+ *	+ *
Alanine	3.79	−	+ *	+ *
Caffeine	3.29	+	+ *	+ *
Kynurenine	4.14	−	+ *	+ *
Lactate	4.11	−	+ *	+ *
Hippurate	7.62	+	+	+ *
Tryptophan	7.18	+	+ *	+ *
Trigonelline	8.81	−	+ *	+ *
Succinate	2.40	+	+ *	+ *
Citrate	2.54	+ *	+ *	+
Hypotaurine	2.60	+ *	+ *	+ *
Taurine	2.69	+ *	+ *	+ *
Melatonin	2.93	+ *	+ *	+ *
Serotonin	3.11	+ *	+ *	+

* *p* < 0.05 with Student’s *t*-test. +/− indicates urinary metabolites greater or lower in each group.

**Table 4 nutrients-12-02900-t004:** Relationship between urinary metabolites for SA with “oats and tropical fruits” DP score and food groups.

Urinary Metabolites	Chemical Shift (ppm)	Dietary Pattern Score		Oats		Tropical Fruits	
		r	*p*-Value	r	*p*-Value	r	*p*-Value
Citrate	2.54	0.05	0.83	0.03	0.92	0.03	0.90
Hypotaurine	2.60	0.31	0.19	0.23	0.34	0.12	0.62
Taurine	2.69	0.14	0.56	0.13	0.58	−0.09	0.70
Melatonin	2.93	0.37	0.11	0.47	0.04 *	0.13	0.60
Serotonin	3.11	0.23	0.32	0.48	0.04 *	0.10	0.70

r = correlation, * *p* < 0.05 with Pearson correlation. Adjusted for age, gender, race, smoking status, body mass index, energy intake, marital status, and education level.

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
