# Peer review of "Urine Untargeted Metabolomic Profiling Is Associated with the Dietary Pattern of Successful Aging among Malaysian Elderly"

_nutrients, 2020, doi:10.3390/nu12102900_

Round 1
Reviewer 1 Report
Dr. Suzana Shahar and colleagues presented in this manuscript a 1D NMR-base metabolomics study of the association between dietary pattern and aging as part of a multiphase longitudinal study. The methods largely followed common consensus and the conclusion seemed to be supported by experimental results. It is suitable to the scope of the Nutrients journal and should be interested to its readership.
I noticed a few drawbacks and minor issues for the authors to revise or comment on.
(1) Line 144, please specify the concentration of phosphate buffer. Was it sufficient to ensure all samples have final pH of 7.4? Chemical shifts of certain metabolites are very sensitive to pH change. Also, Line 142-147 should be merged into 2.2 sample preparation.
(2) Line 100, a relaxation of 2 s is little too short. For absolute quantitation using presat 1D 1H NMR, > 5 s is generally recommended. It is probably not realistic to redo all the experiments, but please be mindful in the future.
(3) To make this manuscript standalone, please briefly summarize the related information of Phase 2 in the method section, e.g. how was dietary controlled for the subjects? What major differences were noticed for different age, gender, etc. and why these factors are insignificant in the current study (Line 141 Models were adjusted...elaborate how?)
(4) Figure 4 labels are overlapping and cannot be read. Either label them better in the plot and remove the legend, or keep the legend and remove all the labels in the plot. Same for Figure 5.
(5) Figure 4 and 5, what are "$M42.DA"??? Please clarify in the figure and text.
(6) For database query, I assume that the frequency domain spectra were used before binning was applied. What was the chemical shift cutoff? How many returns? Were current metabolites identified based only on the highest score or with additional selection criteria? As authors pointed out, 2D and 3D NMR is becoming a gold standard in NMR-based metabolomics.
(7) Check carefully with the writing. e.g Line 37 the first sentence, NMR and mass spec are typically listed together; it is inappropriate to list mass spec vs LCMC due to affiliation relation. Also, there are multiple grammar errors and typos. Need a watchful eye.
Reviewer 2 Report
Study study by Fakhruddin et al sought to extend earler work by the same group which idenfied dietary patterns associated with successful aging (SA) from usual aging (UA) and mild cognitive impairment (MCI). The goal of the present study was to identify urinary metabolomics patterns of successful aging and to correlate these metabolites to specific dietary patterns. The overall findings are interesting and align with earlier work related to biological aging in the plasma metabolome (although this work is not cited). The primary concerns are related to the lack of validation of the metabolites, which were identified by untargeted metabolomics, lack of control for differences in renal function (and thus metabolite excretion). Given lack of key control/validation data, this study should be regarded as preliminary and requires further validation before any definitive conclusions can be made.
Major Comments
- Molecules detected in urine are technically considered “catabolites” because they are the final end-product of the breakdown (catabolism) of molecules that are sent to the kidney for excretion. This should be acknowledged in the discussion when talking about future studies involving other sample sources (e.g., plasma) as the plasma may express different metabolites that are part of the same pathways.
- Urinary metabolite concentrations are typically normalized to urine creatinine concentration to account for differences in GFR among subjects. This is important as those exhibiting successful aging may have better renal function than those with usual aging or MCI and thus any differences in urinary metabolites may be due to kidney function and not differences in food intake/metabolism. This study did not measure actual metabolite concentrations (i.e., no targeted/validation by LCMS was performed) and the abundances reported were not adjusted for renal function.
- Based on the results, it seems that the metabolites reported were only assessed for their association with “oats and tropical fruit” dietary pattern (?) Were other dietary patterns looked at? Based on line 56 of the introduction, I was expecting a more comprehensive analysis linking these untargeted urinary metabolites to multiple dietary patterns, similar to the analysis that this group reported in their previous publication (Ref #8). Based on the Pearson correlation values presented in Table 3, Oats and Tropical Fruits only explain about 25% of the total variance in Melatonin and Serotonin. Are these metabolites more strongly associated with other dietary patterns? This information would be useful when evaluating the utility of urinary metabolomics to predict healhy dietary patterns.
- Line 307: I disagree with the statement that that melatonin and serotonin have been “validated” as biomarkers of the “oats food group”…their abundance has been correlated with this food group; however, a true validation study should measure the actual concentrations of these metabolites (e.g., using targeted metabolomics with external standards) and confirm that they are sensitive to changes in diet (e.g., controlled feeding studies).
- More detail regarding the identification of the 23 metabolites in Table 1 is needed. Untargeted approaches often yield large numbers of metabolites on the order of 100-1000’s of metabolites. The authors should provide more detail on the total number of H-NMR peaks detected and the methods used to arrive at the final list of 23 metabolites. Of the peaks detected, how many could be confidently identified vs. how many were unidentified? How many were endogenous vs. synthetic (e.g., food additives, drug metabolites)?
- There is no mention of any correction for a false discovery rate (FDR) which is OK if properly justified (i.e., in an exploratory, hypotheses generating analysis) but this does not appear to be the purpose of the present study.
- The significant metabolites identified were not confirmed using targeted approaches and rely soley on previously identified chemical shifts.
- Line 139: Again, these models should be corrected for differences in renal function (estimated GFR) which is likely available in the TUA dataset.
- Line 46-47: Several recent papers on metabolomics have reported metabolites in plasma that are associated with indicators of human healthspan and/or the rate of biological aging. These papers should be cited and worked into the introduction as they offer potential insight into plasma-based markers of successful aging:
-
- Johnson et al. Clinical Science. 2018 132:1765-1777 PMID: 29914938
- Johnson et al. GeroScience. 2019. 41:895-906 PMID: 31707594 (including the finding that plasma citrate is indicative of lower biological age)
- Robinson et al., Aging Cell. 2020. 19(6): e13149. PMID: 32363781 (including the finding that tryptophan metabolism is associated with biological age)
Minor Comments
- Title should have the word “is” after the word “profiling”
- Line 25: the word “proven” is too strong here and should be replaced with “assessed” or something softer.
- Figure 5: It would be helpful to add an asterisk (*) above/below bars that represent significant metabolites for east of interpretation by the reader.
- Figure 6 should contain a y-axis.
- The manuscript should be sent for English language editing to correct multiple grammatical errors.
Reviewer 3 Report
Dear Authors,
thank You so much for the manuscript entitled "Urine untargeted metabolomic profiling associated with the dietary pattern of successuful aging among Malaysian elderly". I read it with interest.
Here are my comments and suggestions :
MAJOR COMMENTS :
a) A clear definition of what You mean of "successful aging" seemed appropriate and necessary. Successful aging, in fact, is not only related to cognitive perfomances, as seemed to be reading Your study population and study design (lines 79-80) A table with description of the details of the aging groups classification was mandatory.
b) as You know, mild cognitive impairment (MCI) can be reversible or stable over the years. Only a percentage of older persons with MCI goes toward dementia. In Your manuscript, it was unclear wether there was a relationship between identified metabolites/probable metabolic pathways and the progression risk MCI-dementia.
c) You already published data and research studies on the same topic. How did You avoid a possible autoplagiarism ?
MINOR COMMENTS :
- 1H-NMR (line 37) is for ? As You know, when formula/acronym is reported for the first time, You have to specify what it means. Similarly, in line 47 : successful aging (SA) and in line 51 : hypotheses towards SA.
- Lines from 314 to 320 should be removed. Relationship between sleep pattern and Your results was not a goal of Your research study.
- In previous articles, You used the same NMR spectroscopy approach. Why did not You use a more powerful spectroscope, since You already Knew the limits of the spectoscope You continued to use ?
- References no. 26 and no. 28 are the same.
Round 2
Reviewer 2 Report
The authors have addressed most of my comments; however, I still have some concerns that should be specifically addressed in the text of the manuscript to help the reader evaluate the strength of the results.
- The authors state in their response to my original comment (Points 2 and 8) that the focus of their study is neurodegenerative diseases and not renal function. This may be true, but differences in renal function between groups may still affect the concentration of metabolites detected in urine. Because the authors do not have any data regarding the renal function of their subjects, it is impossible to normalize the data; however, this point should at least be acknowledged in the discussion as a potential limitation and described as a possible alternative explanation for the findings.
- There is no mention in the discussion of the need to confirm the concentration of significant metabolites using targeted approaches in the future. The authors mention this in their response to my original comment but they need to explain this to the reader. They do a nice job of describing the rationale and benefit of using an un-targeted approach but should also discuss the downsides.
- I suggest re-naming their "Successful Aging" group to "Successful Cognitive Aging". As Reviewer 3 points out, there are many ways to define successful aging. Under the description of the MCI group in Table 1, I suggest changing the final bullet to "Preserved global cognitive function" as "global function" implies overall functional status. Also in Table 1 "Subjective memory complain" should be "complaint"
- It should be noted in the discussion that this study was cross-sectional and that future prospective studies are needed to confirm a causal link between this dietary pattern and the biomarkers of successful aging.
- Relatedly, the authors use terms like "increased" and "decreased" in the abstract and throughout the paper to describe differences among groups (e.g., abstract line 26; Table 3). These terms imply a "change" over time in metabolite levels, which is not the case since these are cross-sectional data. I suggest replacing such words with words like "greater" or "lower" when comparing between two groups.
Reviewer 3 Report
All my comments and suggestion were satisfatorily met.
Author Response
Thank you for your kind consideration.